# The Impact of Comorbidities and Obesity on the Severity and Outcome of COVID-19 in Hospitalized Patients—A Retrospective Study in a Hungarian Hospital

**DOI:** 10.3390/ijerph20021372

**Published:** 2023-01-12

**Authors:** Éva Nagy, Viktória Cseh, István Barcs, Endre Ludwig

**Affiliations:** 1Schools of PhD Studies, Semmelweis University, 1085 Budapest, Hungary; 2Bajcsy-Zsilinszky Hospital and Outpatient Clinic, 1106 Budapest, Hungary; 3Department of Internal Medicine and Hematology, Division of Infectology, Semmelweis University, 1088 Budapest, Hungary; 4National Institute of Hematology and Infectious Diseases, South Pest Central Hospital, 1097 Budapest, Hungary

**Keywords:** SARS-CoV-2, COVID-19, obesity, comorbidities, mortality, ICU admission, disease severity, Hungary

## Abstract

Patients with comorbidities and obesity are more likely to be hospitalized with coronavirus disease 2019 (COVID-19), to have a higher incidence of severe pneumonia and to also show higher mortality rates. Between 15 March 2020 and 31 December 2021, a retrospective, single-center, observational study was conducted among patients requiring hospitalization for COVID-19 infection. Our aim was to investigate the impact of comorbidities and lifestyle risk factors on mortality, the need for intensive care unit (ICU) admission and the severity of the disease among these patients. Our results demonstrated that comorbidities and obesity increased the risk for all investigated endpoints. Age over 65 years and male sex were identified as independent risk factors, and cardiovascular diseases, cancer, endocrine and metabolic diseases, chronic kidney disease and obesity were identified as significant risk factors. Obesity was found to be the most significant risk factor, associated with considerable odds of COVID-19 mortality and the need for ICU admission in the under-65 age group (aOR: 2.95; *p* < 0.001 and aOR: 3.49, *p* < 0.001). In our study, risk factors that increased mortality and morbidity among hospitalized patients were identified. Detailed information on such factors may support therapeutic decision making, the proper targeting of vaccination campaigns and the effective overall management of the COVID-19 epidemic, hence reducing the burden on the healthcare system.

## 1. Introduction

The novel coronavirus (nCoV) outbreak caused by the severe acute respiratory syndrome coronavirus 2 (SARS-CoV-2) virus originated in China and was declared a pandemic by the World Health Organization on 11 March 2020 [1]. In the first two years of the epidemic, more than 1,800,000 people were infected in Hungary, of whom more than 44,000 died from complications caused by COVID-19. More than 30,000 of these deaths were recorded in 2021 [2]. 

Several studies aimed at identifying risk factors that influence the progression of SARS-CoV-2 infection. According to a joint statement from the US Food and Drug Administration (FDA) and the National Institutes of Health (NIH), the top-five risk factors for the development of severe coronavirus disease are advanced age (65+), obesity (BMI > 30), type 2 diabetes mellitus, hypertension and other cardiovascular diseases, and chronic lung disease [3], which has been confirmed by other studies [4,5,6].

Hypertension has been associated with increased mortality, severe disease manifestations, the development of acute respiratory distress syndrome (ARDS), the need for ICU admission and disease progression in patients with coronavirus disease [7]. Some studies have found that cardiovascular diseases co-occurring with COVID-19 increase the odds of severity and death (4.02-fold and 6.34-fold, respectively) [8]. Diabetes mellitus, also considered endemic, is a significant risk factor regarding disease severity, increased hospital admission rates and adverse outcomes [8,9,10]. At the start of the pandemic in Italy, more than two-thirds of COVID-19-related deaths were observed in diabetic patients [11].

Risk factors relating to lifestyle, obesity and high BMI (>30) particularly increase the odds of severe COVID-19 manifestation [3]. The prevalence of overweight and obesity is increasing worldwide, reaching epidemic proportions [12]. Obesity increases the odds of several noncommunicable chronic diseases [13,14,15,16] and has also been cited as a risk factor for communicable diseases, particularly since the H1N1 flu pandemic in 2009 [17,18,19]. Obesity inhibits the proper function of the diaphragm, thus complicating the use of respiratory devices, and diminishes the immune response to viral infections [17]. In obese individuals, elevated concentrations of macrophages, cytokines, and proinflammatory adipokines were observed, resulting in a chronic inflammatory state [5]. Kass et al. found a significant inverse relationship between age and BMI, with younger individuals hospitalized with COVID-19 being more likely to be obese [20]. The authors concluded that, in populations with a higher prevalence of obesity, younger age groups may be more affected by the disease [20]. Obesity has also been mentioned increasingly often as an independent risk factor for the disease, rendering the obese population a high-risk group for the SARS-CoV-2 epidemic [5,20,21].

According to the Hungarian Central Statistical Office (KSH), hypertension was one of the most common health problems in the adult Hungarian population in 2019 [22]. The high incidence of cardiovascular diseases is also a major public health problem, accounting for 33% of deaths [23]. Another of the most common chronic diseases is diabetes mellitus, which affects more than 1 million adults in Hungary [22]. Obesity, one of the significant public health problems of our time, also severely affects Hungary. According to KSH data in Hungary, 39.3% of the 18–34 age group and 63.5% of the 35–64 age group were overweight or obese in 2019 [24]. 

Our study aimed at assessing the prevalence of comorbidities and risk factors among SARS-CoV-2-infected patients in a Hungarian hospital and investigating the impact of these on the course and outcome of COVID-19. According to our hypothesis, the presence of underlying conditions and lifestyle risk factors increases the odds of the manifestation of severe and critical SARS-CoV-2 infection, the development of a condition requiring ICU admission and a fatal outcome. 

## 2. Materials and Methods

### 2.1. Study Design and Setting

The single-center study was conducted at the Bajcsy-Zsilinszky Hospital (Budapest, Hungary), from 15 March 2020 to 31 December 2021, involving patients with laboratory-confirmed SARS-CoV-2 infection. This facility has been a designated COVID-19 hospital since 26 March 2020; however, the first patient with a microbiologically confirmed SARS-CoV-2 infection was only diagnosed on 15 March 2020 at the hospital’s Internal Medicine Department. The hospital’s Infection Control Department had been notified of all SARS-CoV-2 patients since the beginning of the pandemic. Patient data had been collected in a surveillance database. This database contained basic demographic information, as well as data on the severity of infection, comorbidities documented on admission and the outcome of infection. The database served as the basis for our retrospective study, with the medical records of all patients reviewed and the data suitable for the study selected. The retrospective data collection from the medical system was approved by the Institutional Committee of Research Ethics of the hospital. The study involved patients over 18 years of age with laboratory-confirmed SARS-CoV-2 infection and requiring hospitalization due to the severity of their illness. Laboratory-confirmed infection refers to the positive results of real-time polymerase chain reaction (rt-PCR) or in vitro immunochromatography (rapid antigen detection) assays for the antigen of the viral nucleocapsid protein. Starting from October 2020, rapid antigen detection was used to diagnose virus infections. The rt-PCR tests were performed by accredited laboratories. The rapid antigen tests were performed and documented by physicians and emergency service professionals. 

### 2.2. Participants and Study Size

Only cases of acute SARS-CoV-2 infection were included in the scope of the study. Patients who had recovered from a coronavirus infection but were hospitalized with post-COVID-19 symptoms were excluded. Those with infection detected within 14 days were considered acute COVID-19 patients. If a patient had more than one episode of infection in the study period, only the first episode of infection or the one requiring hospitalization was included in the study. In the majority of these cases, admission was justified by moderate or severe COVID-19 manifestation or some underlying disease. After cleaning the data and applying the exclusion criteria, 2873 inpatients with SARS-CoV-2 infection could be included in the study database. In Hungary, the first four waves of the COVID-19 epidemic took place between 4 March 2020 and 31 December 2021. We included these four waves in the scope of the study, processing data from patients treated in our hospital. The epidemic waves varied in terms of severity and numbers of patients treated, with 970 and 1903 patients treated in 2020 and in 2021, respectively.

In Hungary, the SARS-CoV-2 vaccination campaigns started in 2021, with millions of people receiving the vaccine. In 2021, a total of 399 people, having previously received full immunization, were admitted to our hospital with SARS-CoV-2 infection. Patients were considered immunized at the time of infection if they had been properly immunized by being vaccinated the number of times and with doses as prescribed by the Summary of Product Characteristics of the given SARS-CoV-2 vaccine. As the vaccine has been available to the general population since 2021, its effectiveness was examined by separately analyzing the data of patients diagnosed in 2021.

### 2.3. Variables and Data Sources

Data from patients hospitalized for SARS-CoV-2 infection were recorded in a Microsoft Excel 2016 spreadsheet. To perform detailed analyses, the population was divided into four age groups: 18–39 years (Age Group 1), 40–64 years (Age Group 2), 65–74 years (Age Group 3) and 75+ (Age Group 4).

Data were collected for each patient by reviewing their respective Emergency Department admission outpatient files, disease course reports by the COVID-19 Department, discharge reports and radiology results. In addition to demographic data, comorbidities, radiology results, days of care, need for ICU admission, vaccination status, and disease severity and outcome were also recorded for the patients. Regarding comorbidities, data for common diseases (such as hypertension and diabetes) and other chronic diseases were collected individually and by disease group. Records on the occurrence of chronic diseases and comorbidities relied partly on patient self-reports and partly on previous medical records. In some cases, chronic diseases previously not detected were diagnosed during hospitalization. Among the risk factors, obesity as well as the lack thereof was also recorded. Data on body weight were not available for BMI calculation, so the nutritional status as described in the hospital admission history was used instead. The nutritional status of patients was assessed by the physician admitting them in the Emergency Department of the Hospital. Due to the increased demand for emergency care as a result of the SARS-CoV-2 epidemic, it was not possible to record the necessary data for BMI calculation; hence, on admission, patients were classified into one of five nutritional groups according to the physical examination performed by the physician and the patient’s parameters as follows: 1. underweight, 2. normal weight, 3. overweight, 4. obese, 5. severely obese. Nutritional status was recorded in the medical records of patients and was then used to classify patients as non-obese (categories 1 and 2) or obese (categories 2–4) during retrospective data collection. Information on other risk factors, such as chronic alcoholism or smoking, was not available for all patients in their respective medical histories; therefore, the effect of these factors was not investigated, lacking the necessary data. Data collection did not include the monitoring of laboratory parameters or the therapies applied.

To assess the outcome of the disease, the procedures of the National Public Health Centre valid at that time were used. Patients were classified as cured according to the criteria set out in these procedures. When recording mortality due to complications, the clinical and pathological diagnosis of death were considered as COVID-19-related mortality by the treating physician and pathologist. To determine the disease severity, the criteria set out in the Therapeutic Manual published by the Minister of Human Resources were applied [25]. 

### 2.4. Outcome Data

The primary endpoint of the study was mortality due to complications from COVID-19. The need for ICU ventilatory support and the degree of severity were considered as additional endpoints. For the latter, patients were classified into five severity categories based on the criteria in the Therapeutic Manual: asymptomatic, mildly symptomatic infection, moderate, severe and critical, where the severity of disease progression was established on a scale of 0 to 4. 

### 2.5. Statistical Methods

Descriptive statistical analysis was performed to express continuous and categorical variables as percentages, as well as providing means, medians, standard deviation and interquartile ranges. In the analytical analyses, the positive probabilities of the occurrence of linear-type variables were modeled using multivariate logistic regression, while scalar-type variables were modeled using linear regression. The regression models were constructed using backward elimination. After elimination, the final model was compared with the initial model. The parameters of the significant variables did not change significantly so, in order to provide more information, the initial model with all the studied parameters was included in the paper rather than the final model received after elimination. Accordingly, our results also demonstrate which independent variables did not have a significant effect on the dependent variables in the study. The chi-square test was applied to individually examine the relationships between nominal variables. For the logistic regression model, goodness-of-fit was tested using the Hosmer–Lemeshow test. To determine the degree of risk, odds ratios with a 95% confidence interval were calculated. Two-tailed α values below 0.05 were considered statistically significant.

Data were collected in Microsoft Excel 2016 and statistical analyses were performed using SPSS Statistics V22 (IBM, New York, NY, USA). The presentation of results followed the Strengthening the Reporting of Observational Studies in Epidemiology (STROBE) guidelines [26]. 

## 3. Results

### 3.1. Patient Characteristics and Descriptive Data

The descriptive epidemiological characterization of the study is summarized in Table A1. The gender distribution was even, with 49.6% (1425) of the subjects being male. The gender distribution was the most uneven in Age Group 4 (75+), where 64.81% of the patients were women. The median age for the entire population was 69 ± 15.57 (58–79) years.

The patients were divided into four age groups, with the highest number of patients belonging to Age Group 4 (75+) (1023 persons, corresponding to 35.6%). Age Group 1 included the lowest number of patients (167, 5.8%). The largest proportion of the total study population had an infection with a moderate course (1064, 37.03%), while 31.19% of the patients developed severe disease and 11.24% of them were in a critical condition. Pneumonia was confirmed in 1940 patients, most of whom were described with CT-confirmed multilobar involvement, which is typical for COVID-19.

During the study period, 358 patients (12.46%) were treated in the ICU. The age groups that required ICU care and respiratory support in the highest numbers were Age Group 2 (40–64 years) and Age Group 3 (65–74 years) (140 patients corresponding to 15.57% and 131 patients to 16.7%, respectively). It was also found that 10.78% of the young adult age group required an elevated level of care. 

Regarding the outcome of the disease, 51.3% of the total study population (1474 patients) left the COVID-19 ward cured. A total of 636 patients (22.14%) returned to their homes in good general condition prior to discharge, either because of their mild disease or because they were transferred to another institution according to the referral policy valid at that time. In their case, no information is available on their further condition or the outcome of the disease. Of the 2873 SARS-CoV-2 infected patients, 763 (26.56%) died, of whom 227 patients were admitted to ICU. Of the deceased, 52.5% belonged to Age Group 4 (75+), i.e., the age group with the highest mortality rate (39.2%). The median age of the deceased was 75 ± 12.25 (67–83) years.

#### 3.1.1. COVID-19 Hospital Days

The average number of days spent in the COVID-19 ward varied between 8 and 13 days, depending on the age group and type of care. In the total population, the average number of days in COVID-19 care was 12.47 ± 8.77 (7–15) days, which increased in direct proportion with age. For Age Group 4 (75+), the average number of days was 13.69 ± 10.03 (7–18). For patients requiring ICU treatment, the average number of hospital days was 10.38 ± 9.06 (4–13.25), with the highest values demonstrated for Age Group 2 (40–64 years) (11.06 ± 10.14).

#### 3.1.2. The Prevalence of Comorbidities and Obesity

In the study population, an average number of 2.4 underlying diseases [2.36 ± 1.48 (1–3)] was calculated per patient, with hypertension showing the highest prevalence (66.55%), followed by other cardiovascular diseases (44.8%) and diabetes mellitus (30.18%); obesity was also present in a high proportion (26.8%). There were only 365 patients (12.7%) who had no known comorbidity at the time of the SARS-CoV-2 infection. 

In Age Group 1, hypertension and diabetes were the most common comorbidities affecting 14.37% each, but the proportion of obese patients was even higher (20.36%). In the other age groups, hypertension was the most common comorbidity, with a prevalence of over 70% in Age Group 3 (65–74) and over 80% in Age Group 4 (75+). The prevalence of cardiovascular disease increased with age, with 65% of people in Age Group 4 (75+) having a history of other cardiovascular diseases. The obesity rate was highest in Age Group 2 (31.48%) and lowest in Age Group 4 (20.23%). 

### 3.2. Main Results

#### 3.2.1. Mortality from Complications of COVID-19

In the overall study population, the odds of a fatal outcome increased with age, with significantly higher odds of death in patients over 65 years of age (OR: 3.1 [CI: 2.55–3.78] *p* < 0.001). Male sex was also identified as an independent risk factor for COVID-19 mortality, where the age-adjusted odds ratio was 1.55 ([CI: 1.30–1.86] *p* < 0.001). In terms of mortality, vaccination was a significant protective factor (*p* = 0.025) and also reduced the odds of developing a critical condition requiring ventilation (*p* = 0.034) (Table 1 and Figure 1). 

A multivariate regression model was used to identify risk factors for mortality, where the moderating effects of underlying diseases and obesity were adjusted for sex, age and vaccination status. The presence and number of underlying diseases significantly influenced mortality (aOR: 2.37 [CI: 1.55–3.66] *p* < 0.001 and aOR: 1.18 [CI: 1.10–1.26] *p* < 0.001, respectively). The odds of mortality were significantly increased by cardiovascular disease (*p* = 0.04), cancer (*p* < 0.001), chronic kidney disease (*p* = 0.001) and obesity (*p* < 0.001) at the time of infection (Table 1 and Figure 1).

The age groups of patients under and over 65 years of age were compared for the risk factors affecting mortality. For those aged below 65 (Age Groups 1 and 2), significant risk factors included cancer (*p* < 0.001), obesity (*p* < 0.001) and the presence of other underlying diseases (*p* < 0.001), whereas male sex, cardiovascular disease and chronic kidney disease did not have a statistically significant effect. For those aged 65 years and over (Age Groups 3 and 4), cardiovascular disease (*p* = 0.003), cancer (*p* = 0.002) and chronic kidney disease (*p* = 0.004) increased the odds of death, but obesity was not a significant risk factor in this group (Table 2).

Looking in more detail at the effect of obesity, the age groups showed different results for mortality from complications. The effect of obesity was tested in three logistic regression models. The most pronounced effect was identified in Age Group 1, where the odds ratio adjusted for age, sex, immunization and comorbidities showed that obesity increased the odds of death more than 14-fold (*p* = 0.028) compared to non-obese patients. The same value is 2.84 (*p* < 0.001) for Age Group 2, while in Age Groups 3 and 4 no significant correlation between obesity and COVID-19 mortality could be found (Table 3).

#### 3.2.2. The Need for ICU Admission

In the study period, the severity of their condition justified ICU admission for 12.46% of the patients. In the overall population, the odds of ICU admission did not significantly increase with age; however, they were significantly affected by sex, being nearly twice as high for men (*p* < 0.001). The presence of at least one comorbidity significantly increased the odds of needing ventilatory support compared to those with no known comorbidity at the time of infection (*p* = 0.004). Obesity and the presence of endocrine and metabolic disease increased the odds of the need for increased ventilatory support in patients with coronavirus infection (*p* < 0.001 and *p* = 0.029, respectively; Table 1 and Figure 1). 

In Age Groups 1 and 2 (under 65 years), male sex and obesity were significant risk factors (*p* = 0.007 and *p* < 0.001, respectively), while the presence of cardiovascular disease positively influenced mortality (*p* = 0.011). In Age Groups 3 and 4 (over 65 years), obesity and male sex were also found to be the only statistically significant risk factors (*p* < 0.001 and *p* = 0.001, respectively). The odds of the need for ICU admission decreased with age (*p* < 0.001). Vaccination against SARS-CoV-2 was also a significant protective factor (*p* = 0.017; Table 2).

#### 3.2.3. Severity of the Disease

The severity of the disease course was measured on a scale of 0–4, with 0 corresponding to asymptomatic and 4 to critical. The average severity score was 2.2 [2.21 ± 1.13 (2–3)], corresponding to moderate severity. Gender was significantly associated with severity, with male patients scoring higher by 0.37 on the severity scale than female patients (*p* < 0.001). Obesity increased severity scores by 0.56 (*p* < 0.001), and the presence of endocrine and metabolic disease by 0.14 (*p* = 0.041). Patients with cardiovascular disease at the time of infection had a lower severity score (*p* < 0.001) compared to those without cardiovascular disease (Table 1). 

Being overweight significantly increased the severity of the disease course in Age Groups 1 and 2 (under 65 years) (*p* < 0.001); male patients also showed higher severity scores compared to female patients (*p* < 0.001) and vaccination against SARS-CoV-2 was identified as a protective factor (*p* = 0.002). A linear negative association was found between the presence of cardiovascular disease or cancer and severity (*p* < 0.001 and *p* = 0.026, respectively). In Age Groups 3 and 4 (over 65 years), male sex, obesity and endocrine disease increased severity (*p* < 0.001, *p* < 0.001 and *p* = 0.037, respectively); however, severity scores were significantly lower in the case of patients with cardiovascular or gastrointestinal disease (*p* < 0.001 and *p* = 0.033; Table 2). 

#### 3.2.4. Impact of SARS-CoV-2 Vaccination

In Hungary, vaccination against COVID-19 was available to the general population from 2021. In 2021, 1903 patients were hospitalized with coronavirus infection, 399 (20.97%) of whom had been vaccinated prior to infection. In the total study sample, a significant protective effect of vaccination was confirmed regarding mortality and need for ICU treatment (*p* = 0.027 and *p* = 0.002, respectively). For those aged below 65, vaccination reduced the degree of criticality (*p* < 0.001). In Age Groups 1 and 2 (over 65 years), vaccination proved to be a significant protective factor for all parameters assessed, reducing the odds of hospitalization (*p* = 0.002), the need for ICU admission (*p* = 0.005) and the degree of criticality (*p* = 0.008; Table 4).

## 4. Discussion

The hypothesis formulated in our objectives, i.e., that the presence of underlying diseases and obesity would increase mortality from complications, the need for ventilatory support and the severity of the disease, was confirmed by the present study. Age over 65 years was found to be a significant risk factor for mortality and the need for ICU admission, with male sex being an independent risk factor when considering all endpoints. Vaccination against SARS-CoV-2 was identified as a significant protective factor for each parameter assessed.

Several Hungarian studies have investigated the epidemiology, clinical features, and efficacy of therapies and vaccination against SARS-CoV-2 infection [27,28,29,30,31,32,33,34,35,36,37,38,39,40,41,42,43,44,45]. However, to our knowledge, none of these involved the assessment of the impact of risk factors in a large, hospitalized population of COVID-19 patients. 

More than half of the COVID-19 patients admitted to hospital (62.9%) were over 65 years of age and only 167 young adults under 40 (5.8%) required hospitalization, confirming the fact that older people, especially those over 65 years of age, are more susceptible to infection due to comorbidities on the one hand, and reduced immunity to a viral infection on the other hand. 

We examined the moderating effect of comorbidities and obesity after adjusting for already identified risk factors (age and male sex) and protective factors (vaccination). Having a comorbidity was a risk factor in itself, but the risk of mortality increased with the number of underlying diseases.

We also examined the risk effect of recorded comorbidities individually. In agreement with several studies on similar topics, hypertension was the most common comorbidity in the overall population and patients over 40. However, in contrast to other studies [5,6,7,8], it was not found to be a significant risk factor in the present study sample. Other cardiovascular diseases also affected a significant proportion of patients, particularly in the over-65 age groups. Their high prevalence in the Hungarian population is well known, with cardiovascular diseases being among the leading causes of death. As also indicated by other studies [5,6,8], cardiovascular diseases increased the risk of death among patients hospitalized for COVID-19 but analysis by age group showed that this effect was only significant in the over-65 age group. The risk effect of diabetes mellitus could not be statistically confirmed in the present study; nevertheless, it was the third most common comorbidity and predominant in all age groups. Respiratory disease showed higher odds ratios at all endpoints, but this was not found to be statistically significant. Endocrine and metabolic diseases increased the need for ICU admission and, in the case of the over-65 age group, the severity of morbidity. Cancer and chronic kidney disease were risk factors for mortality, the latter again statistically significant in the over-65 age group.

Obesity was shown to increase disease mortality and the severity of morbidity. Pathophysiological causes and mechanisms mentioned in the introduction [46,47,48,49,50,51,52,53,54,55,56,57] also play a role in this phenomenon. In our study, the effects of obesity on mortality and morbidity were confirmed by unexpected results. Of all the parameters examined, obesity had the most significant effect on the outcome variables. The obesity rate was highest in the 40–64 age group, but more than 20% of young patients were also obese. Obese SARS-CoV-2-infected patients had a worse disease prognosis and, besides age and male sex, obesity was the only risk factor that significantly increased morbidity and mortality at all endpoints and, therefore, it is considered the most significant risk factor based on the results of this study. In the total study sample, obesity increased the risk of death from complications more than one and a half times; it also increased the need for ventilatory support almost threefold and disease progression by 0.5 severity points compared to the non-obese group. As for the results of age group-specific analyses, it should be noted that, while the negative effect of obesity on morbidity could not be proven for older patients, it still significantly increased the need for ICU ventilation and the severity of the disease in patients over 65. In the case of elderly patients, the effect of obesity on mortality was less pronounced, with advanced age and comorbidities common in the elderly being more important risk factors over 65 years. For those aged below 65, obesity increased the odds of needing ICU treatment nearly threefold, but also had a high impact on those of death and severity. Accordingly, the risk effect of obesity was analyzed further, broken down by age group. In the youngest and oldest age groups (18–39 and ≥75, respectively), obesity rates were the same but, while obesity significantly increased the risk of mortality in young patients, the association was not significant in the oldest age group. In our cohort, in the young adult population aged 18–39 years, obesity increased mortality by more than 14-fold compared to non-obese young patients. Obesity also increased the risk of death nearly threefold in Age Group 2. This finding correlates with the inverse association between age and BMI shown by Kass et al. [20]. Due to the low disease incidence and low mortality in the 18–39 age group, the statistical power of the high-risk value found by the study is low, yet it highlights the high mortality risk in adult obese patients under 40 years of age. In Hungary, more than half of the adult people were overweight or obese in 2019 according to KSH data [24], which underlines the importance of our results.

The effectiveness of COVID-19 vaccination was confirmed by the present study. The complete vaccination series significantly reduced mortality from complications and the need for ventilatory support among patients treated in the hospital. An important finding of the study was the confirmation of the significant protective effect of the SARS-CoV-2 vaccine for all endpoints in patients aged 65 years and older in regression models adjusted for comorbidities, obesity, age and sex. Regarding this effect of vaccination, our results confirm the findings by Vokó et al. who studied a sample representative at national level and found SARS-CoV-2 vaccines to be highly effective in preventing death from complications [43,44].

The study has several limitations. As all the subjects involved in the study were patients of the same hospital, the results obtained are not representative of the Hungarian population. Data collection and research were conducted among hospitalized COVID-19 patients; thus, underlying diseases and risk factors that might have influenced the development of a disease status that required hospitalization could not be included in its scope. The study and statistical analysis focused on the progression and outcome of COVID-19 in hospitalized patients. The inclusion of underlying conditions in the anamnesis was sometimes based on patient self-report, so some comorbidities may not have been diagnosed. Due to the lower number of cases in the 18–39 age group, the values obtained are of limited statistical value compared to other age groups. The coronavirus vaccine was not yet available in 2020, so the protective effect of vaccination could only be tested in patients admitted to hospital in 2021. The prevalence of obesity may be underestimated and the related risk based on BMI values could not be assessed due to the lack of BMI data. The main parameter to establish the level of obesity is BMI. However, the anthropometric data to calculate the index were not available, rendering more precise statistical calculations regarding obesity impossible. It should also be noted that people with low muscle mass may have an abnormal body fat percentage even by a normal BMI. Also, higher than normal BMI may be due to high muscle mass in addition to high physical activity. Accordingly, conducting a physical examination as well as establishing nutritional status based on the patient’s parameters may sometimes be preferable to BMI determination. Furthermore, the most suitable method for adequately determining the accumulation of the most dangerous fat type, known as abdominal or visceral fat, is physical examination, and therefore, in our opinion, the body type described by the physician is professionally acceptable for determining nutritional status and classifying patients into obese/non-obese groups. In addition to obesity, other lifestyle risk factors, smoking and chronic alcoholism, in particular, may often be present and influence the outcome of infection, but it was not possible to assess their impact due to the lack of information. Despite these limitations, our research promotes the identification of statistically detectable predisposing factors. The strengths of our study are the long study period and large cohort, which compensate for the disadvantages of the retrospective methodology. 

## 5. Conclusions

Reducing susceptibility by managing chronic diseases, eliminating lifestyle-related risk factors and increasing physical activity is of paramount importance to prevent severe disease progression and mortality. A significant proportion of the Hungarian population suffers from chronic diseases and/or has a high BMI. Hence, informing and educating the affected populations are important to promote the uptake of booster vaccines and reduce risk factors. In addition to effectively managing epidemics in the community, emphasis should also be placed on providing access to healthcare services to the extent that chronic disease management can be effective and continuous. Untreated chronic diseases (diabetes, hypertension, endocrine diseases, etc.) or a tumor detected late may have detrimental effects on health but, in the case of SARS-CoV-2 infection, they may also contribute to COVID-19 progression. 

Our results highlight obesity as a significant risk factor for severe infection and COVID-19-related mortality regardless of age; therefore, the use of proven and effective pharmacological and nonpharmacological prevention methods is essential in this particular case. It should be noted here that obesity and the presence of comorbidities proven to be statistically significant risk factors should be taken into account when making decisions on, e.g., the need for hospitalization or the selection and proper scheduling of antiviral therapies. Based on the results of this study, we recommend the accurate description of anthropometric data in medical histories to facilitate accurate BMI calculation as well as phenotyping obesity, thus allowing a more accurate and detailed risk assessment regarding the effects of obesity. Obese individuals should be included among the clinically vulnerable groups, should COVID-19 or any other virus attacking the respiratory system cause new pandemics in the future. 

The basic immunization of high-risk groups and the administration of booster vaccines could reduce the number of severely or critically ill patients requiring hospitalization, which in turn could reduce the burden on healthcare systems and diminish mortality among patients infected by SARS-CoV-2.

## Figures and Tables

**Figure 1 ijerph-20-01372-f001:**
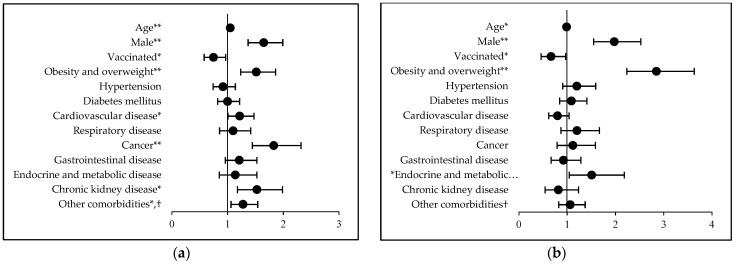
Forest plot graph representing the impacts of comorbidities and obesity on COVID-19-related mortality and the need for ICU treatment. (**a**) Association between comorbid diseases, obesity and death from SARS-CoV-2. (**b**) Association between comorbid diseases, obesity and the need for ICU admission. * Indicates *p* < 0.05 and ** indicates *p* < 0.01. Error bars indicate the 95% confidence interval for each correlation. ^†^ Other comorbidities: neurological and mental health conditions, hematologic diseases, autoimmune diseases, BPH and other urological disorders, rheumatological and orthopedic conditions, etc.

**Table 1 ijerph-20-01372-t001:** Associations between comorbid diseases, obesity, fatal outcome, the need for ICU admission and severity scores of SARS-CoV-2, adjusted for age, sex and vaccination status.

	aOR [95 % CI]		B [95 % CI]
Variables	Model 1	Model 2	Model 3
Age	1.05 ** [1.04–1.06]	0.99 * [0.98–1.00]	0.00 * [0.00–0.01]
Male	1.65 ** [1.37–1.99]	1.98 ** [1.55–2.53]	0.37 ** [0.28–0.45]
Vaccination	0.75 * [0.58–0.96]	0.67 * [0.46–0.97]	−0.11 [−0.23–0.00]
Obesity and overweight	1.52 ** [1.24–1.86]	2.85 ** [2.24–3.63]	0.56 ** [0.46–0.65]
Hypertension	0.92 [0.74–1.14]	1.20 [0.91–1.59]	0.00 [−0.10–0.10]
Diabetes mellitus	1.00 [0.82–1.22]	1.09 [0.84–1.41]	0.03 [−0.06–0.12]
Cardiovascular disease	1.22 * [1.01–1.47]	0.80 [0.62–1.04]	−0.35 ** [−0.43–−0.26]
Respiratory disease	1.10 [0.86–1.41]	1.20 [0.87–1.67]	0.08 [−0.04–0.20]
Cancer	1.83 ** [1.44–2.32]	1.12 [0.79–1.59]	−0.09 [−0.21–0.03]
Gastrointestinal disease	1.21 [0.96–1.52]	0.92 [0.67–1.28]	−0.11 [−0.22–0.00]
Endocrine and metabolic disease	1.14 [0.85–1.52]	1.51 * [1.04–2.18]	0.14 * [0.01–0.28]
Chronic kidney disease	1.53 * [1.18–1.98]	0.82 [0.54–1.24]	0.01 [−0.13–0.14]
Other comorbidities	1.28 * [1.06–1.54]	1.06 [0.83–1.37]	−0.05 [−0.14–0.03]

aOR—adjusted odds ratio; B—unstandardized coefficients. Model 1: association between comorbid diseases, obesity and death from SARS-CoV-2. Model 2: association between comorbid diseases, obesity and the need for ICU admission. Model 3: association between comorbid diseases, obesity and the severity score of COVID-19. The model correlations between Models 1 and 2 were calculated by logistic regression and Model 3 was calculated by linear regression, and each correlation was adjusted for age, sex and vaccination; results were significant at *p* < 0.05. Values in square brackets indicate the 95% confidence interval for each correlation; * *p* < 0.05; ** *p* < 0.01.

**Table 2 ijerph-20-01372-t002:** Associations between age, sex, vaccination, comorbid diseases, obesity and death, need for ICU admission and severity score of SARS-CoV-2 in patient age groups under 65 years (Age Groups 1 and 2) and over 65 years (Groups 3 and 4).

	aOR [95 % CI]				B [95 % CI]	
Variables	Model 1		Model 2		Model 3	
	18–64	≥65	18–64	≥65	18–64	≥65
Age	1.06 ** [1.03–1.08]	1.05 ** [1.03–1.06]	1.02 [1.00–1.04]	0.92 ** [0.90–0.94]	0.02 ** [0.01–0.03]	−0.01 [−0.02–0.00]
Male	1.47 [0.98–2.20]	1.72 ** [1.39–2.14]	1.72 * [1.16–2.54]	1.77 * [1.28–2.46]	0.26 ** [0.12–0.40]	0.35 ** [0.24–0.45]
Vaccination	0.76 [0.41–1.41]	0.76 [0.57–1.00]	0.98 [0.55–1.73]	0.53 * [0.32–0.89]	−0.34 * [−0.57–−0.12]	−0.01 [−0.14–0.13]
Hypertension	1.02 [0.67–1.54]	0.84 [0.65–1.09]	1.22 [0.82–1.82]	0.95 [0.64–1.39]	0.06 [−0.09–0.22]	−0.10 [−0.22–0.03]
Diabetes mellitus	1.16 [0.74–1.80]	0.97 [0.77–1.2]	1.07 [0.70–1.65]	1.04 [0.75–1.45]	0.07 [−0.24–0.11]	0.09 [−0.02–0.19]
Cardiovascular disease	0.66 [0.42–1.03]	1.39 * [1.12–1.72]	0.54 * [0.34–0.87]	1.05 [0.76–1.45]	−0.63 ** [−0.8–−0.45]	−0.21 ** [−0.31–−0.11]
Respiratory disease	0.68 [0.37–1.26]	1.21 [0.92–1.61]	0.92 [0.51–1.63]	1.32 [0.88–1.98]	0.05 [−0.17–0.27]	0.12 [−0.02–0.25]
Cancer	2.95 ** [1.80–4.82]	1.54 * [1.17–2.02]	1.37 [0.79–2.37]	0.85 [0.53–1.36]	−0.25 * [−0.47–−0.03]	−0.07 [−0.20–0.07]
Gastrointestinal disease	1.25 [0.74–2.10]	1.18 [0.91–1.53]	0.94 [0.55–1.61]	0.85 [0.56–1.31]	−0.08 [−0.29–0.12]	−0.14 * [−0.27–−0.01]
Endocrine and metabolic disease	1.08 [0.57–2.03]	1.12 [0.80–1.56]	1.50 [0.83–2.7]	1.48 [0.91–2.42]	0.05 [−0.2–0.3]	0.17 * [0.01–0.33]
Chronic kidney disease	2.01 [1.00–4.07]	1.52 * [1.14–2.01]	0.83 [0.35–1.97]	0.91 [0.56–1.47]	0.17 [−0.15–0.49]	−0.01 [−0.15–0.13]
Other comorbidities	2.32 ** [1.55–3.47]	1.09 [0.88–1.34]	1.39 [0.92–2.10]	0.92 [0.67–1.28]	−0.12 [−0.28–0.05]	−0.01 [−0.11–0.09]
Obesity and overweight	2.95 ** [1.98–4.39]	1.14 [0.89–1.46]	3.49 ** [2.40–5.09]	2.08 ** [1.49–2.90]	0.73 ** [0.57–0.88]	0.40 ** [0.28–0.52]

aOR—adjusted odds ratio; B—unstandardized coefficients. Model 1: associations between comorbid diseases, obesity and death from SARS-CoV-2. Model 2: associations between comorbid diseases, obesity and the need for ICU admission. Model 3: associations between comorbid diseases, obesity and the severity score of COVID-19. The model correlations Model 1 and 2 were calculated by logistic regression, Model 3 was calculated by linear regression, and each correlation was adjusted for age, sex and vaccination; results were significant at *p* < 0.05. Values in square brackets indicate the 95% confidence interval for each correlation; * *p* < 0.05 and ** *p* < 0.01.

**Table 3 ijerph-20-01372-t003:** Effect of obesity and overweight on death from COVID-19, broken down according to age groups.

Age Groups	Prevalence of Obesity (n/%)	aOR [95% CI]		
		Model A	Model B	Model C
Total	770 (26.80%)	1.47 ** [1.21–1.79]	1.47 ** [1.21–1.79]	1.52 ** [1.24–1.86]
18–39	34 (20.36%)	6.9 * [1.54–30.85]	7.87 * [1.57–39.46]	14.83 * [1.34–164.51]
40–64	283 (31.48%)	2.67 ** [1.83–3.90]	2.69 ** [1.85–3.93]	2.84 ** [1.88–4.30]
65–74	246 (31.38%)	1.02 [0.72–1.45]	1.03 [0.72–1.46]	1.16 [0.80–1.68]
≥75	207 (20.23%)	1.20 [0.87–1.65]	1.20 [0.87–1.65]	1.16 [0.83–1.61]

aOR—adjusted odds ratio. Associations between obesity and death from SARS-CoV-2 in the total population and broken down into age groups in different logistic regression models. Model A: adjusted for age and sex. Model B: adjusted for age, sex and vaccination. Model C: adjusted for age, sex, vaccination and comorbidities. Correlations were calculated by logistic regression; results were significant at *p* < 0.05. Values in square brackets indicate the 95% confidence interval for each correlation. * *p* < 0.05; ** *p* < 0.01.

**Table 4 ijerph-20-01372-t004:** Impact of SARS-CoV-2 vaccination on COVID-19 patients in 2021 (n = 1903).

	aOR [95% CI]		B [95% CI]
	Model 1	Model 2	Model 3
Total population	0.73 * [0.56–0.97]	0.47 * [0.29–0.76]	−0.07 [−0.2–0.06]
18–64	0.70 [0.37–1.33]	0.99 [0.55–1.78]	−0.52 ** [−0.73–−0.30]
≥65	0.61 * [0.45–0.83]	0.47 * [0.27–0.80]	−0.31 * [−0.54–−0.08]

aOR—adjusted odds ratio; B—unstandardized coefficients. Model 1: association between vaccination and death from SARS-CoV-2. Model 2: association between vaccination and the need for ICU admission. Model 3: association between vaccination and the severity score of COVID-19. The model correlations between Models 1 and 2 were calculated by logistic regression and Model 3 was calculated by linear regression, and each correlation was adjusted for age, sex, obesity and comorbidities. Values in square brackets indicate the 95% confidence interval for each correlation. * Indicates *p* < 0.05; ** indicates *p* < 0.01.

## Data Availability

Raw data used in the current study are available upon request to the corresponding author.

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
