# Peer review of "The Impact of Comorbidities and Obesity on the Severity and Outcome of COVID-19 in Hospitalized Patients—A Retrospective Study in a Hungarian Hospital"

_ijerph, 2023, doi:10.3390/ijerph20021372_

Round 1

Reviewer 1 Report

The manuscript requires some corrections and has the following shortcomings:

1.      The main parameter for determining obesity is BMI, which is not specified in this manuscript, obesity is determined by the assessment of a health professional. It is necessary to describe in more detail the criteria for classifying patients into the obese or non-obese group.

2.      The results are not clearly shown and are repeated in the text, figure and tables (for example Table 2 and lines 234-250). It is necessary to choose one of the describing methods for each result.

3.      Table A1 and Table 2 need to be formatted so that they are more transparent for reading the results 

Reviewer 2 Report

General comment

The study " The Impact of Comorbidities and Obesity on the Severity and Outcome of COVID-19 in Hospitalized Patients – A Retrospective Study in a Hungarian Hospital " is a potential interesting paper that address an important issue: Comorbidities and risk factors that increase the risk of ICU and death in COVID-19 patients.  The work is well written but there are formal and content aspects that require editorial review

Major comments

The study has some important limitations. All patients are admitted to the same hospital, vaccination began in 2021 (2020 patients did not have the opportunity to be vaccinated) and the main independent variable (overweight-obesity) is not correctly explained as it has been collected. These three aspects should be addressed in Methods, Results and Discussion.

Minor comments

1) Review the last paragraph of the Introduction. Reduce length and better write the objective of the study.

2) Methods:  better explain study design and patient selection.

3) Better explain how the relationship of the dependent variables (ICU, Death and severity) and the main independent variable (overweight-obesity) and the methods to build the regression models are analyzed. the gravity model contributes very little and could be eliminated.

4) there is an error on page 5, line 24: Age group 65+?  and in the first column of table 3 (0-64?)

5) In the first paragraph of the Discussion, assess the main result of the study

6) calculate the effect of the vaccine with the patients of 2021 better the bibliographic review and better discuss its implications

Round 2

Reviewer 2 Report

I have reviewed the article and the authors' response and I think it can be accepted for publication